# Role of Hepatocyte Growth Regulators in Liver Regeneration

**DOI:** 10.3390/cells12020208

**Published:** 2023-01-04

**Authors:** Mitsutoshi Kimura, Hajime Moteki, Masahiko Ogihara

**Affiliations:** Department of Clinical Pharmacology, Faculty of Pharmaceutical Sciences, Josai University, 1-1, Keyakidai, Sakado City 350-0295, Japan

**Keywords:** cytokine, growth factor, signaling pathway, hepatocyte proliferation

## Abstract

We have studied whether growth factors, cytokines, hormones, neurotransmitters, and local hormones (autacoids) promote the proliferation of hepatic parenchymal cells (i.e., hepatocytes) using in vitro primary cultured hepatocytes. The indicators used for this purpose include changes in DNA synthesis activity, nuclear number, cell number, cell cycle, and gene expression. In addition, the intracellular signaling pathways from the plasma membrane receptors to the nucleus have been examined in detail for representative growth-promoting factors that have been found to promote DNA synthesis and cell proliferation of hepatocytes. In examining intracellular signaling pathways, the effects of specific inhibitors of presumed signaling factors involved have been pharmacologically confirmed, and the phosphorylation activities of the signaling factors (e.g., RTK, ERK, mTOR, and p70 S6K) have been evaluated. As a result, it has been found that there are many factors that promote the proliferation of hepatocytes (e.g., HGF, EGF, TGF-α, IL-1β, TNF-α, insulin, growth hormone (GH), prostaglandin (PG)), and serotonin (5-HT)), while there are very few factors (e.g., TGF-β1 and glucocorticoids) that inhibit the effects of growth-promoting factors. We have also found that 5-HT and GH promote the proliferation of hepatocytes via different autocrine factors (e.g., TGF-α and IGF-I, respectively). Using primary cultured hepatocytes, it will be possible to further study the molecular and cellular aspects of liver regeneration.

## 1. Introduction

The liver is the central organ of metabolism and is responsible for the homeostasis of the body by metabolizing carbohydrates, proteins, lipids, and other substances. Furthermore, detoxification and regeneration are characteristic functions of the liver. These abilities are thought to have been acquired during the long evolutionary process to protect animals from catastrophic damage to liver tissues caused by food toxins (toxic chemicals) [1].

The mechanisms of liver regeneration have been studied for many years, and numerous reports have been published. There are also excellent review articles integrating those results [1,2,3,4,5,6]. The 70% partial hepatic resection of rodent livers (PHx) has been widely used as a research model for liver regeneration. For example, when approximately 70% of the liver of an anesthetized rat is surgically removed, the residual liver spontaneously and rapidly initiates cell division and proliferation, regenerating to the volume and weight of the original organ, and this response is automatically terminated [1,2]. In this animal model, the only signals for the initiation of cell proliferation are local wounding and tissue loss, making this a simpler experimental system than models of liver regeneration from viral infection or chemical-induced liver injury. Nevertheless, in vivo experimental systems have inherent difficulties in examining the detailed mechanisms of liver regeneration after PHx, because many factors are involved in a single event.

Therefore, we used a simpler in vitro primary cultured hepatocyte system to examine the actions of individual hepatocyte growth factors (and candidate substances) and their intracellular signaling pathways. Primary cultured hepatocytes retain metabolic activity comparable to that in vivo and express many kinds of receptors, each of which is known to be responsive to agonists [3,7]. It is also known that, among the parenchymal (i.e., hepatocytes) and nonparenchymal (i.e., Kupffer, pit, stellate, endothelial, etc.) cells that constitute liver tissues, the hepatocytes initiate proliferation at the earliest stage.

In this review, an in vivo model of liver regeneration is briefly described. In addition, based on our results, the factors that promote or inhibit the proliferation of hepatocytes are classified into their bioactive characteristics, and their respective intracellular signaling pathways are briefly summarized. Then, we will discuss the physiological significance of how each of these factors may play a role in liver regeneration in vivo.

## 2. Animal Models of Liver Regeneration

The first scientific study of liver regeneration was performed by Higgins and Anderson using a PHx animal model in rats [8]. The rodent PHx model involves surgical removal of the left lateral, left medial, and right medial lobes, which is equal to an approximately 70% decrease in the liver mass. This method, although a classical model, is still the best option today. In general, the cycle of liver regeneration after PHx consists of three critical steps: (1) priming, (2) progression, and (3) termination. The DNA synthesis is maximal on day 1, and these three steps are completed in 5 to 7 days, allowing the residual liver to regenerate to its original weight (LW/BW; liver weight/body weight) (Figure 1) [9,10]. This means that, at the cellular level, the hepatic parenchymal cells and other nonparenchymal cells automatically begin proliferating by some molecular mechanisms, activating the cell cycle (G_0_ → G_1_ → S → G_2_ → M → G_0_ phase), which stops when the organ returns to its original size. During the post-PHx, the blood levels of the intracellular enzymes aspartate aminotransferase (AST) and alanine aminotransferase (ALT) increase rapidly due to the destruction of the liver tissue (Figure 1).

The most important question is: what triggers and what terminates liver regeneration after PHx? Moolten and Bucher showed that carotid-to-jugular cross-circulation between hepatectomized and normal rats, via polyethylene cannulas, stimulated the incorporation of ^14^C-thymidine into hepatic DNA in normal partners [11]. Their results provided convincing evidence of a humoral mechanism in liver regeneration after PHx. Using the PHx model, we showed that epidermal growth factor (EGF) induces a significant increase in the LW/BW ratio by increasing 5′-bromo-2′-deoxyuridine incorporation into hepatocyte DNA in the remnant liver on days 2 and 3 after PHx when compared with saline-treated control rats [9]. 

Furthermore, the expression of many hepatocyte genes is reportedly induced within the first hour after PHx. It is also known that noradrenaline, hepatocyte growth factor (HGF), tumor necrosis factor-α (TNF-α), interleukin (IL)-6, serotonin (5-HT), and bile acid levels in the blood also increase after PHx [4]. Therefore, all of these factors could be thought to trigger liver regeneration after PHx.

Although the aforementioned in vivo study of liver regeneration using live animal models has its advantages, there are limitations when studying the intracellular signaling mechanisms in detail. The main reason is that the liver tissue is composed of several different cell groups (i.e., parenchymal and nonparenchymal cells), and the mediators released by each cell affect each other in liver regeneration. Therefore, in the early 1970s, a method was developed to isolate, purify, and culture hepatocytes by using collagenase to perfuse the liver. This technique made it possible to study the intracellular signaling pathways for the proliferation of hepatocytes. As a result, it is now believed that liver regeneration does not initiate and progress by a single humoral factor, but rather by the interaction of a number of growth-promoting factors that exert a vigorous regenerative capacity.

## 3. Isolation, Purification, and Primary Culture of Hepatocytes

The hepatocytes are isolated from normal livers by the two-step in situ collagenase perfusion technique devised by Seglen to facilitate disaggregation of the adult rat liver [12]. The hepatic parenchymal and nonparenchymal cells are separated and purified by repeated low-speed centrifugation. Then, the purified hepatocytes are cultured in a medium containing 5% newborn calf serum for 3 h, and the cells are allowed to adhere to the bottom of collagen-coated culture dishes (at this time, TNF-α and IL-1β, which are considered priming factors, are not added). 

After 3 h of incubation, the culture medium is replaced with a serum-free, defined medium, and the incubation is continued for an arbitrary period of time (0–21 h). Under the serum-free, defined culture conditions, the various effects of growth regulators on hepatocyte proliferation can be observed when different types of growth factors (and candidate substances) and signal-transducing factor inhibitors are added alone or in combination [13].

Under normal physiological conditions, hepatocytes are in the quiescent phase of the cell cycle (G_0_) and have stopped proliferation. However, in our culture system, the cell density at the time of cell seeding is relatively low, and the concentration of dexamethasone is reduced without adding insulin or EGF, which are considered essential for cell survival. By doing so, we have found that the effects of individual growth factors on hepatocyte proliferation can be observed much earlier than before.

Under the above conditions, we also found that the cell cycle already progresses from the G_0_ phase to the G_1_ phase after 3 h of incubation with serum, although the cause is unknown at present (Figure 2) [14].

When target growth factors are added immediately after the change to a serum-free, defined medium, we have observed that the progression from the S phase to the M phase continues after the phosphorylation of the signaling factors, and the number of nuclei (or cells) increases significantly after about 3–4 h (Figure 3A,B).

We also found that there are many factors that promote hepatocyte proliferation, but there are few factors that can strongly inhibit the effects of these hepatocyte growth-promoting factors (Table 1). Recently, we were also able to confirm the hepatocyte proliferation-promoting effects of 5-HT and growth hormone (GH) in culture [14,15,16,17,18]. Therefore, we investigated the associated intracellular signaling pathways and found that these regulatory factors are indirect mitogens that act via the secretion of autocrine factors (transforming growth factor-α (TGF-α), insulin-like growth factor-I (IGF-I), etc.).

## 4. Classification and Characteristics of Hepatocyte Growth Regulators

Mitogens can be defined as factors that promote the proliferation of hepatocytes by themselves, such as EGF, TGF-α, HGF, platelet-derived growth factor (PDGF), and IGF-I. Co-mitogens, by themselves, do not promote hepatocyte proliferation, but, when used in combination with mitogens, the co-mitogens enhance the activity of the mitogens, and these include adrenergic α- and β-agonists and glucagon. In addition, there are many indirect mitogens, such as TNF-α, IL-1β, prostaglandin (PG) E_2_, 5-HT, and GH, which indirectly promote hepatocyte proliferation by secreting autocrine factors. In contrast, some factors, such as transforming growth factor-β1 (TGF-β1) and glucocorticoids, strongly suppress mitogen-induced hepatocyte proliferation (i.e., they are inhibitory factors). The physical properties, receptors, intracellular signaling pathways, and bioactive characteristics of representative hepatocyte growth-promoting and growth-inhibitory factors are briefly summarized below (Table 1).

### 4.1. Mitogens

#### 4.1.1. Direct Mitogens

##### EGF

EGF is a protein with a molecular weight of approximately 6 kDa, and it has three disulfide bonds. EGF binds to the EGF receptor (EGFR) and activates intracellular signaling pathways by phosphorylating the receptor tyrosine kinase (RTK); the EGF receptor is phosphorylated in vivo 30–60 min after PHx [3].

The proliferative effects of EGF on hepatocytes have been examined in culture. The results have shown that EGF alone promotes DNA synthesis and the proliferation of hepatocytes. It has been suggested that RTK, extracellular signal-regulated kinase (ERK), and the mammalian target of rapamycin (mTOR) are involved in the signaling [13]. Noradrenaline and an adrenergic β_2_ receptor agonist, on their own, do not promote hepatocyte proliferation, but these agents enhance the growth-promoting effects of EGF. Thus, there is crosstalk between the adrenergic β_2_ receptor-mediated signaling system and the EGF receptor-mediated signaling system (Figure 4).

##### TGF-α

TGF-α is a protein with a molecular weight of approximately 5.5 kDa, which has high amino acid homology with EGF and binds to EGFR/ErbB1. After receptor dimerization, TGF-α activates RTK, which, in turn, activates the mitogen-activated protein (MAP) kinase cascade via Smad, an adaptor protein, for intracellular signal transduction [35]. TGF-α, a cytokine produced by keratinocytes and various cancer cells, directly promotes DNA synthesis and the proliferation of hepatocytes alone in culture [19]. It is also an autocrine factor secreted by hepatocytes in response to several cytokines (see below).

The effects of TGF-α on hepatocytes have been investigated in culture. The results have shown that TGF-α alone promotes DNA synthesis and the proliferation of hepatocytes and that RTK, phosphatidylinositol-3 kinase (PI3K), ERK, and mTOR are involved in its signal transduction. In addition, adrenergic α_1_ receptor agonists enhance the effects of TGF-α, suggesting crosstalk with adrenergic α_1_ receptor-mediated signaling pathways (Figure 4).

##### HGF

HGF is a protein consisting of 728 amino acids and a heterodimeric structure, with a heavy chain of approximately 60 kDa and a light chain of 3.5 kDa. It is a cytokine produced by stellate cells and endothelial cells in the liver. Its receptor is cMet, which dimerizes and activates the RTK in the intracellular domain to mediate the action of the HGF [36,37,38].

The proliferative effect of HGF on hepatocytes has been examined in culture. The results have shown that HGF alone promotes DNA synthesis and the proliferation of hepatocytes and that RTK, PI3K, ERK, and mTOR are involved in its signaling [20]. We have also found that both adrenergic α_1_ and β_2_ receptor agonists potentiate the growth-promoting effects of HGF by crosstalk (Figure 4).

##### PDGF

PDGF belongs to the PDGF/VEGF family. Its molecular weight is approximately 30 kDa, and its A and B chains form homo- and hetero-dimeric structures (PDGF-AA, PDGF-BB, and PDGF-AB), which activate RTK for intracellular signal transduction [39]. It was first isolated from platelets, but it is also produced by macrophages, vascular endothelial cells, smooth muscle cells, and cancer cells. In addition to PDGF, the platelets also contain TGF-β1, HGF, and 5-HT, which are released during the platelet adhesion and aggregation associated with tissue injury [3]. The released PDGF is thought to trigger a series of responses including inflammation and macrophage, neutrophil, and fibroblast migration.

We investigated the proliferative effects of PDGF on hepatocytes in culture. The results showed that PDGF-BB alone promotes DNA synthesis and proliferation of hepatocytes and that RTK, PI3K, ERK, and mTOR are involved in its signal transduction [21]. In addition, an adrenergic α_1_ receptor agonist enhances the effects of the PDGF-BB, suggesting crosstalk with adrenergic α_1_ receptor-mediated signaling pathways (Figure 4).

##### Insulin

Human insulin is a heterodimer consisting of an A chain of 21 amino acids and a B chain of 30 amino acids, joined by two disulfide bonds. It is produced by pancreatic B cells and can always act on the liver via the portal bloodstream. The receptor for insulin is a built-in tyrosine kinase, and its activation phosphorylates its intracellular substrate, insulin receptor substrate-1 (IRS-1). Subsequently, signals are transmitted to PI3K and protein kinase B, and glucose transporter-4 is translocated to the cell surface. It has also been reported that blood levels of insulin increase soon after PHx [40].

The proliferative effect of insulin on hepatocytes has been said to be a co-mitogen that enhances the action of direct mitogens such as EGF. Therefore, we investigated the effects of insulin in culture. Our results showed that insulin alone promotes DNA synthesis and hepatocyte proliferation and that RTK, PI3K, ERK, and mTOR are involved in its signaling [22,41]. In addition, an adrenergic α_1_ receptor agonist potentiates the action of insulin, suggesting crosstalk with adrenergic α_1_ receptor-mediated signaling pathways (Figure 4).

#### 4.1.2. Co-Mitogens

##### Noradrenaline

Noradrenaline is a sympathetic neurotransmitter with a catecholamine structure. Noradrenaline receptors include adrenergic α and β types, each of which has subtypes. All the receptors for noradrenaline are G-protein-coupled ones. It has been reported that sympathetic hyperactivity due to post-PHx invasion increases the blood levels of noradrenaline in about 1 h [42]. It activates the duodenal Brunner’s gland and stimulates EGF production and HGF expression [43,44].

In primary cultured hepatocytes, noradrenaline alone does not promote the proliferation of hepatocytes, but, through crosstalk, it can enhance the hepatocyte proliferation-promoting effects of EGF [13], TGF-α [19], HGF [20], PDGF [21], and insulin [22] (Figure 4). In addition, in combination with phenylephrine (a selective adrenergic α_1_-receptor agonist), it enhances the hepatocyte-proliferative effects of EGF, HGF, TGF-α, PDGF, and insulin. Metaproterenol (a selective β_2_-receptor agonist) enhances the proliferative action of EGF, IGF-I, and HGF. These results show that noradrenaline exhibits unique regulatory mechanisms for these growth factors.

#### 4.1.3. Indirect Mitogens

##### IL-1β

IL-1β was the first inflammatory cytokine among the ILs to be identified, and there are two types, IL-1α and IL-1β [45]. IL-1β is a protein consisting of 110–140 amino acids. IL-1 receptors are highly homologous to toll-like receptors and activate serine/threonine kinases via adaptor proteins for intracellular signal transduction. IL-1 is rarely detected in normal tissues and is produced and secreted by macrophages and other immune cells activated by infiltrating inflammation. It has proliferative effects on monocytes and granulocytes.

The proliferative effects of IL-1β on hepatocytes have been examined in culture. The results have shown that IL-1β alone promotes DNA synthesis and the proliferation of hepatocytes and that the IL-1β effects are mediated via autocrine secretion of TGF-α from hepatocytes (Figure 5) [24]. IL-1α does not promote the proliferation of hepatocytes.

##### TNF-α

TNF-α is a protein consisting of 157 amino acids and is a potent proinflammatory cytokine similar to IL-1 and IL-6. TNF-α promotes the proliferation of hepatocytes via TNF-α type 1 receptors [46]. The major TNF-α-producing cells in peripheral tissues are macrophages (Kupffer cells in the liver), but it is also released from mast cells. It is involved in the pathogenesis of chronic inflammatory diseases such as Crohn’s disease and rheumatoid arthritis.

We investigated the proliferative effects of TNF-α on hepatocytes in culture. The results showed that TNF-α alone promotes DNA synthesis and the proliferation of hepatocytes and that its effects are mediated via its autocrine secretion of TGF-α from the hepatocytes (Figure 5) [25].

##### PGE_2_ and Prostacyclin (PGI_2_)

Prostaglandins (PGs) are low-molecular-weight local hormones. The receptors for PGs include EP, IP, and TP types, all of which are G protein-coupled. PGs are synthesized under the influence of cyclooxygenase (COX) from arachidonic acid and excised from the plasma membrane by phospholipase A_2_. The physiological process involves constitutive COX-1 expression, while the inflammatory process involves inducible COX-2 expression. Biosynthesized PG products exhibit a variety of physiological effects, but inactivation is relatively rapid. PGs act on cells of the immune system to enhance inflammatory responses (redness, fever, swelling, pain, and dysfunction) by promoting the release of IL-1 and TNF-α. PGI_2_ is produced by vascular endothelial cells, regulates blood flow, and inhibits platelet aggregation. It has been reported that PGs stimulate hepatocyte DNA synthesis in culture [47].

In primary cultured hepatocytes, we investigated the intracellular signaling pathways of PGE_2_ and PGI_2_, which promote DNA synthesis and the proliferation of hepatocytes. As a result, we found that both PGE_2_ and PGI_2_ indirectly promote DNA synthesis and the proliferation of hepatocytes via the secretion of the autocrine factor TGF-α from the hepatocytes (Figure 5) [39,40,48].

##### 5-HT

5-HT is a low-molecular-weight substance with receptor subtypes designated 5-HT_1_ to 5-HT_7_. 5-HT acts as a neurotransmitter in the central nervous system and as a messenger molecule in the periphery. For example, it is released from enterochromaffin (EC) cells in the small intestinal epithelium and acts as a local hormone. 5-HT is stored in platelets and released upon stimulation and is involved in blood coagulation by promoting platelet aggregate formation (hemostatic thrombus). 5-HT has been reported to be involved in the promotion of liver regeneration [49]. That is, thrombocytopenic mice have impaired liver regeneration, and the administration of 5-HT restores liver regeneration. However, since platelets also contain other mediators (PDGF, HGF, etc.) that promote hepatocyte proliferation, it is necessary to examine whether this is a direct or indirect action of 5-HT.

When we examined the direct action of 5-HT on the proliferation of hepatocytes in culture, we found that 5-HT promotes the autocrine secretion of TGF-α from the hepatocytes via the 5-HT_2B_/Gq/phosphoinositide-specific phospholipase C (PLC)/Ca^2+^ pathway. We then found that the secreted TGF-α directly promotes DNA synthesis and hepatocyte proliferation (Figure 5) [14,17,18].

##### GH

GH is a single-chain peptide consisting of 191 amino acids, which is secreted from the anterior pituitary gland. GH induces functional changes in the metabolic capacities of various organs, such as the growth and differentiation of cells and tissues, bone mineralization, and the metabolism of carbohydrates, lipids, and proteins [28]. Its receptor is a Janus kinase 2 (JAK2)-related type. In vivo, GH is known to promote liver regeneration in PHx rats [50].

We examined the effects of GH on the proliferation of hepatocytes and intracellular signaling pathways in culture. GH promotes the autocrine secretion of IGF-I from the hepatocytes via the GH receptor/JAK2/PLC/Ca^2+^ pathway. We then found that the secreted IGF-I directly promotes DNA synthesis and hepatocyte proliferation (Figure 5) [15,16,23].

### 4.2. Hepatocyte Growth Inhibitory Factors

#### 4.2.1. Inhibitory Mitogens

##### TGF-β1

TGF-β1 is a protein with a molecular weight of approximately 2.5 kDa. There are three isoforms of TGF-β: TGF-β1, -β2, and -β3. TGF-β is produced by platelet, bone, placenta, and many cancer cells [31]. The TGF-β receptor is a serine/threonine-type receptor that phosphorylates Smads for intracellular signal transduction. Although it was first identified as a factor that promotes fibroblast transformation, it also inhibits proliferation and induces cell differentiation and apoptosis in many cell types. It acts on epithelial cell proliferation in an inhibitory manner [3].

The effect of TGF-β1 on the hepatocyte proliferation-promoting effects of EGF, HGF, and TGF-α has been examined in primary culture, and they are significantly inhibited at very low doses of TGF-β1 [32].

##### Glucocorticoids

Glucocorticoids are natural steroid hormones produced in the adrenal cortex that include cortisol and cortisone. In contrast, dexamethasone is a synthetic steroid hormone derivative. They cross the plasma membrane, bind to intracellular receptors, and their complexes migrate into the nucleus to regulate gene transcription. For example, one protein produced by steroids is lipocortin, which inhibits the first step in the cleavage of arachidonic acid from the cell membrane lipids. This suppresses PG production and inhibits the inflammatory response of tissues due to trauma or infection.

The effect of dexamethasone on the hepatocyte proliferative effects of EGF, HGF, and TGF-α has been examined in culture, and low doses of dexamethasone inhibit the proliferative effects of these growth factors in a time-dependent manner [19,33,34].

#### 4.2.2. Control of Hepatocyte Proliferation by Cell-to-Cell Contact

It is known that cell density during in vitro culture of hepatocytes affects the action of growth factors [51]. Therefore, we have examined the effects of cell densities at the time of cell seeding on the proliferative actions of several growth factors. As a result, we found that some growth factors do not show their growth-promoting effects when the cell density becomes relatively high (EGF and HGF) [13,20], while others are not affected by cell density (PDGF and insulin) [21,22]. These results indicate that some hepatocyte growth-promoting factors may act in the early stages of liver regeneration, while others exert their effects at a relatively late stage in vivo.

## 5. Discussion

As mentioned earlier, liver regeneration in vivo can be divided into three important phases: (1) priming, (2) progression, and (3) termination. It has been recognized that liver regeneration begins early in each of these phases due to the interplay of numerous regulatory factors [1]. However, studies on the actions of the numerous growth regulators that initiate, progress, and terminate liver regeneration are fragmented, and there has been little comprehensive discussion of the role of these regulators over the natural time course of liver regeneration. Fausto et al. proposed that, during the natural course of liver regeneration, cytokines are initially activated, followed by growth factors and metabolic systems, and that liver regeneration proceeds through the cooperation of each of the three networks of systems [2]. We basically support their view, but we would like to add the following additional perspectives based on the results of our studies.

### 5.1. Noradrenaline as a Co-Mitogen

When the sympathetic nervous system is activated by invasive stress caused by PHx, noradrenaline is rapidly released from the nerve endings. Furthermore, in vivo, it has been reported that noradrenaline induces the early secretion of EGF from the endocrine glands. Noradrenaline can stimulate adrenergic α or β receptors and enhance the hepatocyte proliferative effects of EGF, HGF, PDGF, TGFα, and insulin by crosstalk in culture (Figure 4). That is, the stimulation of the adrenergic α1 receptor enhances the proliferative effects of HGF, PDGF, TGF-α, and insulin. Adrenergic β_2_ receptors enhance the proliferative effects of EGF and HGF separately. Then, it is conceivable that transported endocrine mediators, either locally or somewhat distantly via the bloodstream, serve as signals to enhance the proliferation of hepatocytes at an early stage in vivo (Figure 4) [2,3].

### 5.2. PG and 5-HT as Indirect Mitogens

In liver tissues damaged by PHx, PGs are produced relatively early, and an acute inflammatory response occurs. Immune system cells, such as macrophages and basophils, are activated by promoting the release of mediators (TNF-α and IL-1β) to phagocytose and remove necrotic tissues. In addition, an early hemostatic response occurs at the site of the vascular injury. In other words, platelets adhere and release stored mediators (e.g., 5-HT) to promote the aggregation reaction. The platelets also release their own growth factors, such as HGF and PDGF. PGs and 5-HT, which are involved in inflammation and hemostasis, have been shown to act as indirect growth-promoting factors mediated by autocrine factors in vitro (Figure 5) [17,26,27]. Therefore, the PGs and 5-HT are presumed to play initiator and promoter roles in vivo.

Moreover, we also found that the branched-chain amino acid leucine and vitamin C stimulate hepatocyte proliferation via an autocrine mechanism [29,30]. However, the signaling pathways remain to be elucidated.

### 5.3. TGF-β1 and Glucocorticoids as Inhibitory Mitogens

TGF-β1 strongly inhibits the hepatocyte proliferation-promoting effects of EGF, HGF, and TGF-α in culture [31,32]. In vivo, it has been reported that TGF-β1 increases in the blood very quickly—2–3 h after PHx—and persists for 72 h afterward [3]. However, during the first half of liver regeneration (~3 days), the inhibitory effect of TGF-β1 may not be exerted because of the predominant action of the multiple growth-stimulating factors. Only in the second half of liver regeneration (3 to 7 days), when the actions of many of the growth factors are completed, the TGF-β1 effects become dominant, the LW/BW is restored, and liver regeneration is considered to be complete (Figure 1) [4]. It has been suggested that the activation of adenylate cyclase is involved in the inhibitory effect of TGF-β1 on hepatocyte proliferation in response to TGF-α stimulation in culture. However, the detailed mechanism of action remains to be elucidated [32]. Furthermore, in vitro, dexamethasone strongly inhibits the hepatocyte proliferative effects of EGF, HGF, and TGF-α [19,33,34]. Thus, TGF-β1 and glucocorticoids are presumed to be involved in the termination of liver regeneration in vivo.

## 6. Future Perspectives

Recently, the molecular mechanisms of action of various factors that regulate hepatocyte proliferation in vitro have been gradually elucidated. Based on these results, the in vivo mechanisms of liver regeneration are also becoming better understood. By integrating the results of these basic biological studies, we hope to extract useful insights that can be appropriately applied to several clinical issues (e.g., promotion of liver regeneration after liver cancer resection or living donor liver transplantation) [6].

## Figures and Tables

**Figure 1 cells-12-00208-f001:**
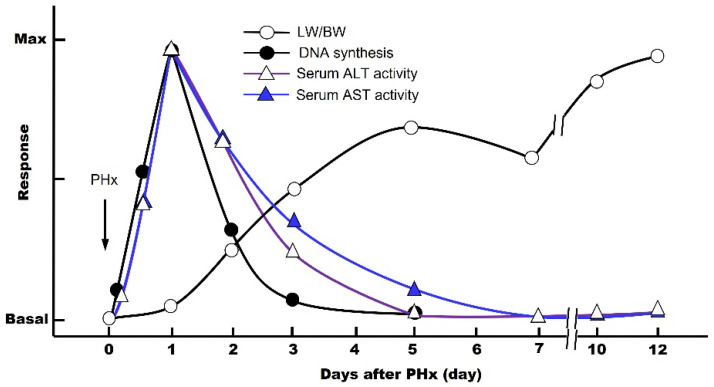
Temporal sequence of events associated with 70% partial hepatectomy (PHx) in vivo. Changes in serum transaminase activity, DNA synthesis activity, and LW/BW ratio.

**Figure 2 cells-12-00208-f002:**
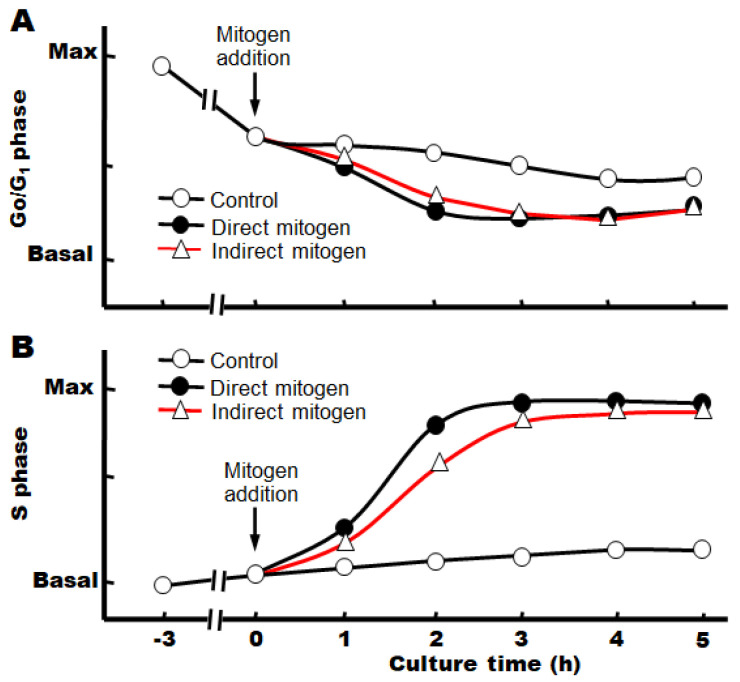
Cell cycle progression stimulated by mitogens in primary cultures. (**A**) Percent of total hepatocyte nuclei in the G_0_/G_1_ phase of the cell cycle. (**B**) Percent of total nuclei in the S phase of the cell cycle.

**Figure 3 cells-12-00208-f003:**
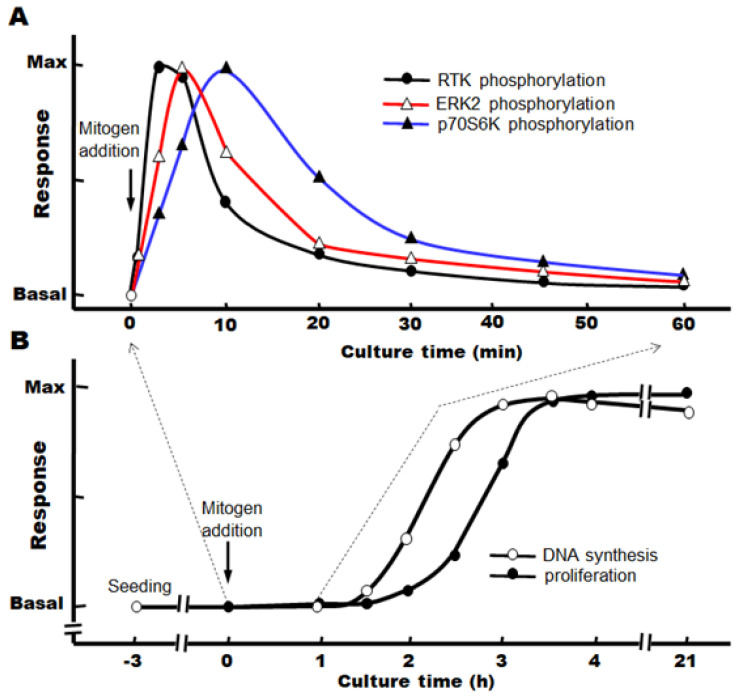
Time course of hepatocyte proliferation stimulated by mitogens in primary cultures. (**A**) Phosphorylation of signaling factors by mitogens, (**B**) DNA synthesis and hepatocyte proliferation induced by mitogens.

**Figure 4 cells-12-00208-f004:**
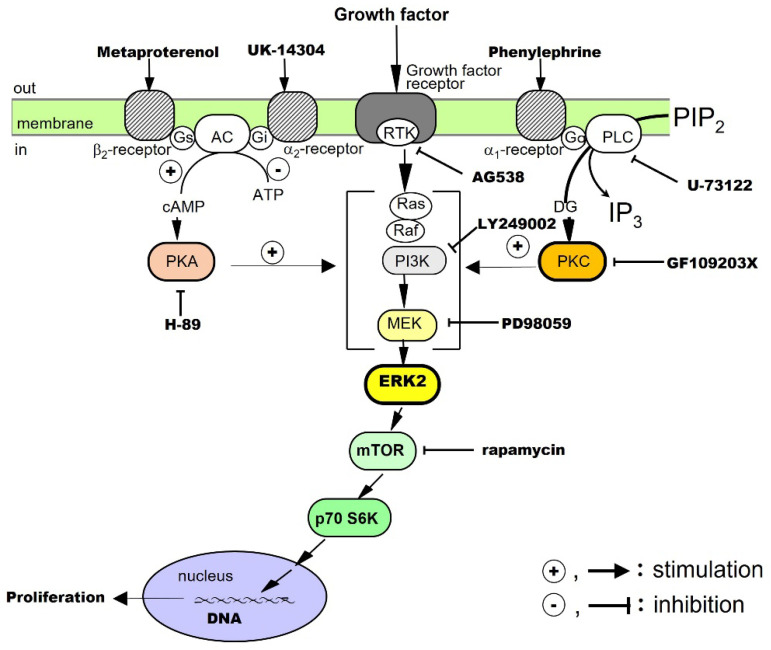
Co-mitogenic effects of α- and β-adrenergic receptor agonists on hepatocyte proliferation stimulated by growth factors.

**Figure 5 cells-12-00208-f005:**
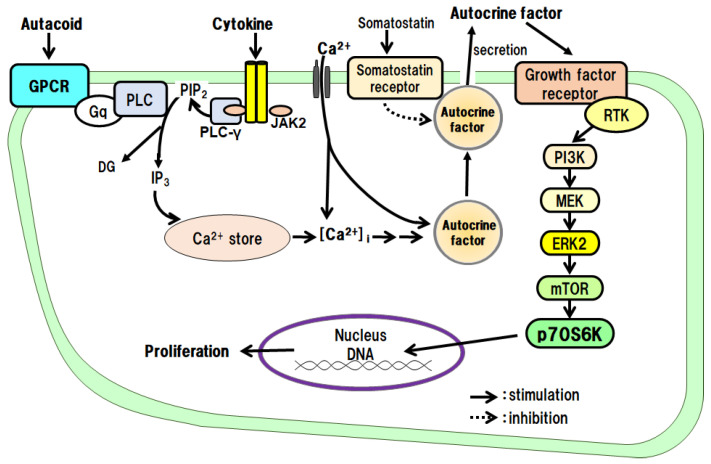
Hepatocyte proliferative effects of autacoids and growth hormone via an autocrine mechanism.

**Table 1 cells-12-00208-t001:** Classification of hepatocyte growth regulators and their intracellular signaling pathways.

Growth-Related Factors	Signaling Pathways	References
1. Mitogen		
(1) Direct: growth factor	RTK-ERK-mTOR-p70S6K	
EGF	RTK-ERK2-mTOR-p70S6K	[13]
HGF, PDGF, TGF-a, insulin, IGF-I	RTK-PI3K-ERK2-mTOR-p70S6K	[19,20,21,22,23]
IGF-II	ERK2-mTOR-p70S6K	[23]
(2) Indirect	GPCR/RTK-JAK-PLC→RTK-ERK2-mTOR-p70S6K	
cytokine: IL-1b, TNF-a	GPCR-PLC→RTK-PI3K-ERK2-mTOR-p70S6K	[24,25]
autacoid: 5-HT, PGE_2_, PGI_2_	GPCR-Gq-PLC→ RTK-PI3K-ERK2-mTOR-p70S6K	[14,17,26,27]
growth hormone	RTK-JAK2-PLC→RTK-PI3K-ERK2-mTOR-p70S6K	[15,16,28]
other: BCAA, vitamin A, C	?→PLC→RTK-PI3K-ERK2-mTOR-p70S6K	[10,29,30]
2. Co-mitogen		
α-adrenoceptor agonists, NA	GPCR-Gq-PLC-IP_3_/DAG-PKC	[19,20,21,23]
β-adrenoceptor agonists, glucagon	GPCR-Gs-AC-cAMP-PKA	[13,22,23]
3. Inhibitory factor		
TGF-β_1_	TGF-β1-R-(Gs)-AC-PKA/RSTK-Smad/ERK-mTOR-p70S6K	[31,32]
glucocorticoids	GR-HRE-transcriptional activation	[19,33,34]

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
