# Peer review of "Role of Hepatocyte Growth Regulators in Liver Regeneration"

_cells, 2023, doi:10.3390/cells12020208_

Round 1

Reviewer 1 Report

The authors provide a comprehensive review on current knowledge regarding the regulation of hepatocyte regeneration.

As expert from the field they did an excellent job and wrote a review worth reading, briefly summarizing the current state-of-the-art. It is well-illustrated.

Ik only have a few minor points:

1) On line 98-99: Perhaps "IL" should be "IL-1" or 'IL-1beta"?

2) In the first column some of the growth factors seem to be on the wrong line. This may be an editing issue. Please align these.

Author Response

Response to reviewers

We thank you for your careful review and constructive suggestions. The original manuscript has been modified according to the reviewers' comments.

Comment #1

Response

  1. We have changed the word IL to IL-1b on p.3, line 117, of the revised manuscript.
  2. We have changed the word make up to constitute on p.2, line 49, of the revised manuscript.

Reviewer 2 Report

With this review, the authors aim to examine what is known about the role of specific regulators in the liver regeneration process. Beyond the purpose (and the title!), the authors poorly explain some mechanims of action and/or take some information for granted, making it difficult for a non-specialist reader to understand. 

Author Response

Response to reviewers

We thank you for your careful review and constructive suggestions. The original manuscript has been modified according to the reviewers' comments.

Comment #2

Response

We have added some explanations so that even the non-expert reader can understand some mechanisms of action.

  1. We have changed the placement of the introduction paragraph on p.1, lines 25-45, and p.2 lines 46-54, of the revised manuscript. Personal references have described on p.4, lines 157-163, of the revised manuscript.
  2. We have described some of the evidence obtained using the PHx model in the text (p.2, lines 70-98, and p.3 lines 99-100, of the revised manuscript) and added a reference to Moolten and Bucher as Ref. 11.
  3. As suggested, we have added the definitions of mitogen, co-mitogen, and indirect mitogen, p.4, lines 192-200, of the revised manuscript.
  4. Figures and a Table are placed at the end of the section.
  5. As suggested, we have divided section 4 into shorter subparagraphs.

We are grateful to the reviewers for bringing these to our notice.

Reviewer 3 Report

The proposed manuscript offers a good overview of the factors involved in liver regeneration.

Although the content appears well-provided, some aspects should be better discussed or summarised in order to make the reading more fluent and easier to understand even for non-liver specialists. Furthermore, the general organisation of the manuscript should in my opinion be revised.

In particular:

Section 1 (Introduction) - The second part of the section seems badly placed. It would be preferable to introduce the topic in a more formal way and without personal references, which I think would be fine if quoted throughout the rest of the paragraphs (particularly from section 3 onwards). Furthermore, the reference to Figure 1 is not necessary at this point in the text.

Section 2 - The authors refer well to the knowledge gained from the in vivo experiments, but the paragraph seems a little too short. Adding a few more important passages (highlighting a chronology of findings) or adding a diagram or table summarising what has been done in animal models to date would be useful and make the manuscript more complete for the reader.

Section 3 - Could the authors conclude the paragraph with a better transition between the possibility of using cell culture (isolated PHHs) and the classification of growth factors? The transition seems too sharp.

Section 4 - Also in this passage, the introduction to the paragraph is too abrupt. It would be preferable to introduce the PHHs and their classification (direct/indirect mitogens, co-mitogens etc.) and then present them one by one. The table can be referenced after such an introduction or at the end of the paragraph, but not at the beginning when the reader is supposed to be a non-expert in the subject. Table detail: format so that the text is more readable; formatting in the middle of the text does not make it easy to read. Another detail: the sub-paragraphs in section 4 should be better spaced out. The text as it is now is heavy.

Please pay attention to the English revision (i.e. page 2 line 54: constituted instead of make up)

Author Response

Response to reviewers

We thank you for your careful review and constructive suggestions. The original manuscript has been modified according to the reviewers' comments.

Comment #3

Response

  1. We have changed the placement of the introduction paragraph on p.1, lines 25-45, and p.2 lines 46-54, of the revised manuscript. Personal references have described on p.4, lines 157-163, of the revised manuscript.
  2. We have described some of the evidence obtained using the PHx model in the text (p.2, lines 70-98, and p.3 lines 99-100, of the revised manuscript) and added a reference to Moolten and Bucher as Ref. 11.
  3. As suggested, we have added the definitions of mitogen, co-mitogen, and indirect mitogen, p.4, lines 192-200, of the revised manuscript.
  4. Figures and a Table are placed at the end of the section.
  5. As suggested, we have divided section 3 into shorter subparagraphs, p.3 lines 112-132, of the revised manuscript.

We are grateful to the reviewers for bringing these to our notice.

Round 2

Reviewer 2 Report

This reviewer can not provide any comments on the revised manuscript. It is not possible to check the changes done since they do not appear as track changes. Furthermore, this reviewer it is not able to find in the revised manuscript the changes done by the authors as suggested in the rebuttal letter: pages and lines cited by the authors in the rebuttal letter do not correspond to what it is shown in the revised manuscript. 

Reviewer 3 Report

The authors revised the manuscript as suggested